# Talking about Health: A Topic Analysis of Narratives from Individuals with Schizophrenia and Other Serious Mental Illnesses

**DOI:** 10.3390/bs12080286

**Published:** 2022-08-13

**Authors:** Tovah Cowan, Zachary B. Rodriguez, Ole Edvard Granrud, Michael D. Masucci, Nancy M. Docherty, Alex S. Cohen

**Affiliations:** 1Department of Psychology, Center for Computation and Technology, Louisiana State University; Baton Rouge, LA 70803, USA; 2Department of Psychology, Kent State University, Kent, OH 44242, USA

**Keywords:** schizophrenia, serious mental illness, health promotion, exercise, eating, hobbies, physical health

## Abstract

Individuals with schizophrenia have higher mortality and shorter lifespans. There are a multitude of factors which create these conditions, but one aspect is worse physical health, particularly cardiovascular and metabolic health. Many interventions to improve the health of individuals with schizophrenia have been created, but on the whole, there has been limited effectiveness in improving quality of life or lifespan. One potential new avenue for inquiry involves a more patient-centric perspective; understanding aspects of physical health most important, and potentially most amenable to change, for individuals based on their life narratives. This study used topic modeling, a type of Natural Language Processing (NLP) on unstructured speech samples from individuals (*n* = 366) with serious mental illness, primarily schizophrenia, in order to extract topics. Speech samples were drawn from three studies collected over a decade in two geographically distinct regions of the United States. Several health-related topics emerged, primarily centered around food, living situation, and lifestyle (e.g., routine, hobbies). The implications of these findings for how individuals with serious mental illness and schizophrenia think about their health, and what may be most effective for future health promotion policies and interventions, are discussed.

## 1. Introduction

It is well established that individuals with schizophrenia and other serious mental illnesses (SMI), such as bipolar disorder, schizoaffective disorder, and major depressive disorder, have higher mortality and reduced life expectancy [1]. Specifically, individuals with schizophrenia live approximately fifteen years less than populations without serious mental illness [2].). This higher mortality is in part due to higher natural causes of death, such as cardiovascular disease, cancer, and other illnesses [1,3,4],with cardiovascular diseases representing the leading cause of death for these individuals [5]. Substantial inquiry has explored the contributory causes in this population which lead to higher negative health outcomes, such as cardiovascular events and other metabolic diseases such as diabetes [6,7]. Many major factors which connect mental health conditions to poor physical health outcomes have been identified, such as side effects of psychiatric medications, lack of access to or engagement in physical activity, lack of access to healthy food and other diet related factors, and substance use, particularly higher rates of nicotine and alcohol use [8,9,10]. Notably, not all studies have found consistent disparities in health behaviors in the serious mental illness population, particularly regarding mixed findings around exercise and cigarette use [11,12,13] though see also [14] for significant disparity in sedentarity. However, the consistent finding of poorer health outcomes has stimulated significant efforts to close the health and life-expectancy gap for individuals with schizophrenia, and SMI more broadly, and improving physical health for these populations is a critical issue.

Unfortunately, after decades of research on health promotion and self-management interventions, there are no indicators that the physical health of individuals with schizophrenia has improved, and, in fact, the mortality gap appears to be widening [4,15] (though see [16] for evidence that in some countries, it has not widened). Many interventions have been focused on improving individuals’ health behaviors with varying effectiveness (e.g., [17,18,19]). While increasing individual health behaviors is undoubtedly an important component of improving the overall health of individuals with schizophrenia, there are several systemic components which must also be addressed, including stigma [20,21], diagnostic overshadowing, where individuals’ physical concerns are ignored in favor of their psychiatric symptoms [22,23,24], and access to health-promoting environments [25,26,27]. Focusing solely on the individual, without understanding the broader context, will likely be insufficient for understanding what is necessary in improving the physical health and reducing the mortality of individuals with schizophrenia. Indeed, new applications of ecological frameworks to schizophrenia symptomatology have underscored the importance of understanding environmental contributions for symptoms [28], and similar thinking may be relevant and required for physical health promotion as well.

Qualitative findings from focus group discussions of individuals with SMI have also emphasized the importance of understanding context. Individuals with SMI who participated in focus groups identified that their social surroundings could be a significant help (e.g., emotional, practical, mutual support from family members or significant others) or barrier to health behavior change [29]. While these effects are not limited to individuals with SMI, they may be compounded by other structural impacts on the social environments of individuals with SMI, e.g., less access to public spaces due to stigma or transportation/safety concerns and more limited social networks [30].

Shrinking the mortality gap between individuals with and without SMI will require intervention on multiple levels, including individual, contextual, and systemic barriers to health, and is an urgent problem. This combination of urgency and complexity highlights the need for effective and efficient interventions. But, to know what will be effective and efficient in the lives of individuals with SMI, their expertise must be included. There has been significant work, often in the context of designing and evaluating behavioral health interventions, on what individuals with schizophrenia see as the barriers and facilitators of their health change (e.g., [26,31,32,33,34]). Individuals with SMI identify substantial concerns around their physical health [33,34]. Several of these studies highlighted how strong patient-care provider relationships, and general social relationships, can support the individual’s health-promoting behaviors and facilitate navigating barriers [26,32,33,34]. On the other hand, social norms around health behaviors, particularly around food, can be a hinderance to making lifestyle changes, as can symptoms such as amotivation or depression [26,34]. They cohere with the quantitative findings on the importance of health behaviors and health promotion for individuals with schizophrenia, and the critical roles of social systems both formal and informal. They also highlight how specific barriers, such as psychiatric symptoms and medication, may be particularly at play for this population [34]. However, these qualitative inquiries are often particularly targeted to a specific intervention, particularly interventions around exercise and diet (though see [33] for a systems-oriented perspective).

This study aims to continue the trajectory of centering individuals’ lived experience and perception of their health and wellbeing, but in a larger and more varied context than is feasible or applicable for qualitative inquiries. Using topic analysis, a Natural Language Processing (NLP) method which allows for the automatic extraction of relevant topics, we aim to explore how individuals with schizophrenia perceive their own physical health behaviors as part of their larger life context. In topic analysis, the words in a transcribed speech sample are clustered according to an algorithm, which has been trained on a large corpus. These topics, consisting of contextually similar clusters of words, can then be qualitatively assigned a name, much in the way one would if doing an exploratory factor analysis. We aimed to identify if individuals with schizophrenia discussed behaviors relevant to physical health in unstructured and untargeted speech samples, and, if so, what kinds of health-relevant topics emerged. In an exploratory manner, we also aimed to identify how individuals themselves valence their own physical health promoting or detrimental behaviors, and what the broader context of those behaviors are. Lastly, we compared whether the frequency of topics differed in the group of individuals with schizophrenia, relative to other serious mental illness diagnoses or controls without a mental illness diagnosis. By exploring these perspectives, we will gain new insights on the aspects of health which are most relevant and most valued for individuals with schizophrenia, informing potential intervention and policy development to improve physical health and reduce mortality for this marginalized community.

## 2. Methods

### 2.1. Participants

This study drew from archival data from three studies. Further details regarding these studies are reported elsewhere [35,36,37], and only details relevant to the current inquiry will be reported here. For participant demographic information, including age, gender distributions, diagnostic status, and racial/ethnic distributions, please see Table 1. Average age of participants was 39.8 years old (*SD* = 10.1, range = 18–67).

### 2.2. Measures

#### 2.2.1. Procedures

In each study, participants were audio recorded while they discussed memories from their lives. See Table 1 for details regarding each study’s procedure. For two of the studies, these were conducted as part of a semi-structured conversation with the research assistant. For one of the studies, this was conducted as part of a monologue for which prompts and probes were provided to the participant before and during the speaking task. For two of the studies, positive, negative, and neutral affectively valenced memories were separately elicited as part of the speaking task instructions and the follow-up probes. For the other study, affective valence was ambiguous with respect to initial instructions, and participants were allowed to explore whatever affectively valenced memories they deemed important. Recordings were hand-transcribed by trained research assistants, and interviewer speech was removed from the samples for this study. After filtering, we had 1482 transcripts from 366 individuals (*M* (*SD*) = 4.04 (2.3), range = 1–8).

#### 2.2.2. Natural Language Processing

Using Python (v3.8) and Natural Language Toolkit Library (NLTK) [38], documents were filtered by length (character length > 50; word length > 10), contractions were expanded, stop-words (e.g., the, an, a, should) were removed, and filtered for alphanumeric characters and phrases (periods, commas, hyphens, and apostrophes were retained). After cleaning and filtering, we had 1481 text documents transcribed from videos (*M* (*SD*) *=* 601 (585), range = 50–2763 words per document).

### 2.3. Topic Analysis

To extract topics, BERTopic [39] was used to identify frequent similar phrases, clustered into topics. We chose BERTopic over the many other clustering algorithms because of its ease of use, interactive visualizations for inspecting topics, and automatic optimization of the number of topics (k). Unlike traditional Latent Dirichlet Allocation methods [40], BERTopic does not require you to set as an input the number of clusters. After dimensionality reduction, it finds dense areas of similar documents in the vector space using HDBScan [41], a density-based clustering algorithm. We used BERTopic library, ngrams between 1 and 3 words, and the ‘all-mpnet-base-v2′, a pre-trained sentence transformer based on Microsoft’s ‘mpnet-base’ that maps sentences and paragraphs to a dense vector space and can be used for tasks like clustering or semantic search. This model was used because it was developed as a sentence and short paragraph encoder, in contrast to models that were trained on longer, multi-paragraph documents. BERTopic then uses a class-based term frequency inverse document frequency (c-TF-IDF) algorithm to identify human-interpretable topics whilst keeping important words in the topic description [39]. C-TF-IDF compares the importance of terms within a cluster and creates easily interpretable term representation: the higher the value is for a term, the more representative it is of its topic. Ultimately, in order to better analyze the potentially large array of topics, BERTopic offers an interactive intertopic distance map and hierarchical clustering for inspecting individual topics [39]. Once an initial overview of the topics becomes available, an automated topic reduction can be performed again.

### 2.4. Data Analysis

We used Chi-squared tests to compare the frequency of topics between the following groups: (a) Individuals with an SMI diagnosis broadly and controls, (b) individuals with schizophrenia, individuals with another SMI diagnosis, and controls, and (c) Black and White participants. These group comparisons were performed using a one-way Chi-squared test on the distribution across all topics.

Topics were identified as “health-related” by evaluating cosine similarity scores between each topic and health-related terms. Cosine similarity is a method for approximating how similar two units of text (words, phrases, sentences, or documents) are using text embeddings. In brief, text embeddings are a vector (numerical) representation of a document, word, or sentence in semantic space, calculated by the probability of text appearing close to each other in natural language. Natural language is often composed of text found on Wikipedia, social media, or other large corpora such as news articles, books, or social media. Cosine similarity scores are calculated by subtracting the cosine distance between two vectors in semantic space from one. The further the words are in semantic space (not similar), the closer their cosine similarity score is to 0. Once topics were identified, we searched resulting topics and terms that had high cosine similarity (>0.5) to health-related terms (i.e., “health”, “exercise”, “smoking”, “alcohol”, “sleep”, and “self-care”). We assigned emotional valence to health-related topics by qualitatively inspecting the top 10 individual words in health-related topics (Table 2). We compared the frequency of health-related topics among groups using a two-way Chi-square test for 7 health-related topics and all other topics.

## 3. Results

### 3.1. What General Topics Emerged?

Topic analysis revealed 42 distinct topics (Table 2; Appendix A). Names were given based on the terms that contributed the most to a topic in reference to their C-TF-IDF weights. Figure 1 illustrates the distribution of all topics and how similarly they cluster with one another. Globally, the distribution of topics was significantly different between individuals with an SMI diagnosis and controls (Chi-squared = 54.124, *p* = 0.004). These differences did not appear to be driven by ethnicity, as the distribution of global topics was not significantly different between racial/ethnic groups (Chi-squared = 30.173, *p* = 0.456).

Several topics were discussed at greater frequency (4, 7, 11, 12) in individuals with an SMI diagnosis compared to the control group (see Figure 2). When considering only individuals with an SMI diagnosis, Topics 4, 5, 15, 28 were discussed at a significantly different frequency in schizophrenia patients than in patients with other SMI diagnoses. These included pertinent terms like food, hobbies, relaxing, work, fun memories, people, job, and medication. Lastly, there were 16 non-health-related topics discussed amongst individuals with an SMI diagnosis that were not discussed in the control group. These topics included terms such as family relationships, work, stress, mental illness, and mental health. While these topics and constituent terms have a qualitative overlap with “health”, they were not included in our top seven health-related terms based on a cosine similarity threshold of 0.5.

### 3.2. What Health Related Topics Emerge and How Are They Emotionally Valenced?

Seven topics were revealed to have health-related themes. These topics and their emotional valences are also listed in Table 2, along with a select few other topics for comparison.

### 3.3. Is Frequency of Health-Related Topics Related to Patient Group?

Health-related topics were discussed in greater frequency by individuals with SMI when compared to controls (Chi-squared = 5.994, *p* = 0.014), but these topics were not discussed differently by individuals with schizophrenia versus other SMI diagnoses (Chi-Squared = 3.841, *p* =0.475). Importantly, three health-related topics were discussed in individuals with schizophrenia, but not individuals with other SMI diagnoses. When comparing individuals with schizophrenia and those with other SMI diagnoses, we found that Topics 1 “hobbies”, 3 “routine”, 10 “scared”, and 13 “religion” were individually discussed at different frequencies.

## 4. Discussion

The primary finding from this inquiry is that, even without a specific health-related prompt, individuals with schizophrenia and SMI broadly identified several health-related topics, and in fact do so more than controls. Specifically, participants discussed food and diet and several lifestyle factors, such as routines, hobbies, and living situations. Interestingly, the relative frequency of health-related topics compared to all other topics was not significantly different between individuals with schizophrenia, individuals with other SMI diagnoses, and controls. However, we find several individual topics that are brought up by individuals with schizophrenia more frequently than individuals with other SMI diagnoses, including several lifestyle factors. The most striking topic to emerge from these unstructured interviews was the first one, which broadly discusses food and diet, and for which the top four contributing words are “food, eat, cook, love.” In planning these exploratory analyses, we considered the possibility that individuals with SMI would discuss health-related topics, but in a manner that did not clearly identify the valence they associate with these behaviors. That was not the case. It is clear that, at least in this sample of individuals with SMI, food is positively valenced. Maintaining that positive relationship with food, rather than a punitive diet framework, will likely be beneficial for long term health promotion. Negatively valenced emotion topics, particularly “scared”, also emerged as health-related topics. Closer examination of the other words in the topic suggests that the health-related component may be “run”, which in this topic could be the activity, or could be something related to escape. In sum, it is clear from these analyses that individuals with SMI, primarily schizophrenia and schizophrenia spectrum disorders, are actively thinking about their health even when not explicitly prompted to, and areas of primary importance to them are food and diet and their general lifestyle and routines.

One potential health-related domain also stands out for its absence in the topics extracted from these speech samples. Even though some of the speech samples targeted negatively valenced memories and experiences the participants have had, there was not a clearly related disability, sickness, or chronic illness topic. There are several possible interpretations to this absence. First, the participants may have had these experiences, but they were not sufficiently salient to be discussed in these interviews. It does appear that many of the negatively valenced topics coalesce around more interpersonal stress or negative experiences like lethargy. These may simply be more common or clearly negative experiences for the individuals in the sample, and therefore the memories which came to mind. It may also be that the participants in our sample, though on average middle aged, had not yet had many significant negative health outcomes to this point. Globally, individuals with schizophrenia live to an average age of 64.7 years old [2], and our sample was approximately two decades younger, and therefore may have not yet developed chronic health conditions which are common in this population, such as diabetes or cardiovascular diseases [1,3]. A third possibility is that it reflects diagnostic overshadowing, or even internalized diagnostic overshadowing. Diagnostic overshadowing typically refers to the phenomenon of individuals with SMI receiving worse physical health care, not least because medical doctors are apt to attribute their concerns to psychiatric symptoms [24,42,43]. It is conceivable that our sample has yet to receive proper diagnosis for disabilities. It is also possible that there has been an aspect of internalized diagnostic overshadowing, analogous to internalization of stigma, where the individuals in our sample have learnt that the most significant contributions to their health and wellbeing are their psychiatric symptoms, and that their physical health symptoms are less relevant.

### Implications for Health Promotion

Our findings have several critical implications for health promotion initiatives for individuals with schizophrenia. First and foremost, they suggest the importance of targeting diet. In the general population, diet has been identified as a key component of improving cardiovascular health [44] and mitigating negative outcomes from diabetes [45]. Given that this is already a topic which is important to individuals with schizophrenia, this may indicate a domain of their life where health promotion initiatives can have significant impact. However, one crucial nuance that our findings suggest is the importance of a positive association with food. In a population where a loss of pleasure (i.e., anhedonia) is a cardinal symptom, the fact that food, eating, and cooking were strongly positively valenced suggests the importance of the hedonic component of food. As such, heavily restrictive diets (i.e., elimination diets) could be particularly ineffective in this population (though these diets are generally ineffective, even in populations without mental health diagnoses [46]). Instead, advice on diet should focus on maintaining cooking and eating as a source of joy through providing culturally appropriate, non-shaming health-promoting additions to the individual’s diet. However, this does not necessarily suggest that the primary strategy should be individualized diet advice, though that may be useful. Instead, it is important to understand the relation between diet, eating, and social context, and interventions need to work on individual, microsystem, and exosystem levels [47]. At the broadest level, policy initiatives such as ensuring the access to plentiful healthy food options [48] and the infrastructure for cooking it, are required—without access no other intervention can succeed. Subsequently, interventions which stimulate the creation or maintenance of supportive attitudes within the individuals’ social network [49] and providing individual diet information and strategies for integrating health-promoting cooking styles will be required to make substantial changes in the health of individuals with schizophrenia. A similar strategy or perspective could be applied to initiatives to increase exercise in this population. Research shows higher rates of sedentarity in individuals with schizophrenia [50], but also the importance of self-determination in motivation towards increased physical activity in individuals with schizophrenia [51]. Our results suggest hobbies are important to individuals with SMI. Increasing the ease and frequency with which individuals can participate in exercise-related hobbies, for example, by increasing access to greenspace or to safe neighborhoods, can increase both physical and mental health for individuals with SMI [52,53,54].

## 5. Limitations and Future Directions

The results of this study should be interpreted considering some limitations. It is to be expected with topic modeling that sample characteristics and task characteristics will have a substantial impact on the topics extracted. Several factors at play in our samples may have influenced the topics. First and foremost, the samples were drawn from studies collected across a decade, and in two different parts of the United States. Replication across different samples will be necessary. Within our samples, we did not find significant differences in the topics extracted based on ethnicity, but future work should expand the samples and the ethnic groups included and explore whether these findings replicate. If these findings do not replicate, it may suggest limits to generalizability, and ways that either policy or intervention ought to be culturally adapted, whether that reflects the broader culture of a geographic region or the culture of a particular ethnic or racial group. Finally, these results reflect the prompts which were provided to the participants, which were not specifically health related, but some of the speech prompts targeted good memories, bad memories, or neutral experiences. Still, health related topics emerged prominently, showing that these are integral to individuals with SMI’s self-narratives.

## 6. Conclusions

Relatively unstructured speech samples from individuals with SMI, primarily schizophrenia, were analyzed using natural language processing and topic modeling to explore what health-related ideas were critical to this population. Topics around food, living situation, and lifestyle (e.g., routine, hobbies) emerged. There was a notable lack of topics specifically relating to ill health. The topics which are primary to individuals with schizophrenia, and how they were discussed, provide information on how to create more effective interventions to improve the health and wellbeing of this population.

## Figures and Tables

**Figure 1 behavsci-12-00286-f001:**
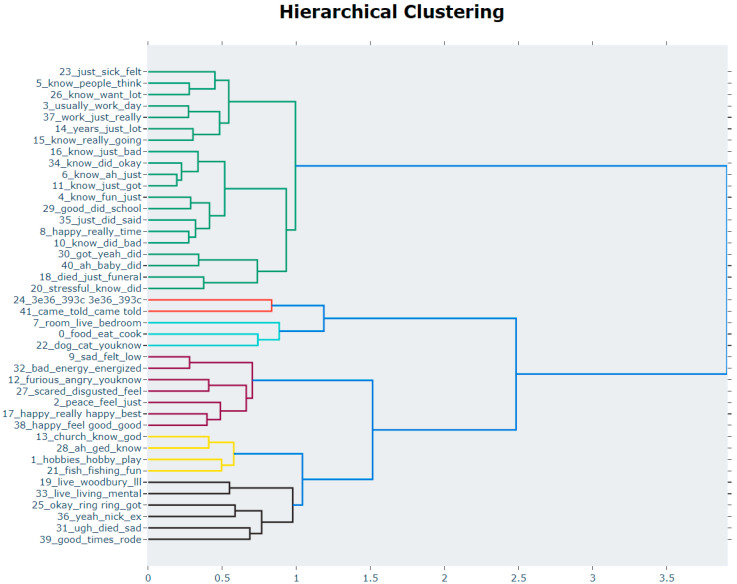
Hierarchical clustering of topics.

**Figure 2 behavsci-12-00286-f002:**
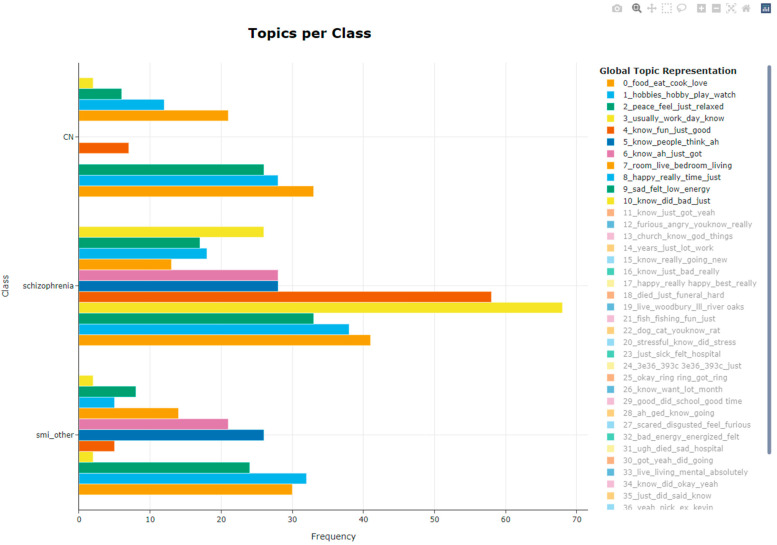
Frequency of topic discussion by diagnostic group. For ease of interpretation, only the first eleven topics are included. Relevant topic names as seen in Table 2 are as follows: Topic 0 = Food; Topic 1 = Hobbies; Topic 2 = Peace; Topic 3 = Routine; Topic 4 = Memories; Topic 5 = People; Topic 6 = Life; Topic 7 = Home; Topic 8 = Happy; Topic 9 = Sad; Topic 10 = Scared; CN = Controls.

**Table 1 behavsci-12-00286-t001:** Demographic and descriptive statistics for studies.

	Study 1	Study 2	Study 3	Total
	(Docherty et al., 2003)	(Cohen et al., 2008)	(Cohen et al., 2014)	
**Sample Size (Schizophrenia Spectrum/Other SMI/Controls)**	141/0/0	0/94/0	43/50/38	184/144/38
**Number of Language Samples (Schizophrenia Spectrum/Other SMI/Controls)**	591/0/0	0/94/0	298/236/247	889/330/247
**Sex (Men/Women)**	68/60	63/21	57/42	188/123
**Race (Black/White/Other)**	23/79/1	53/31/0	41/59/4	117/169/5
**Age (M ± SD)**	37.2 ± 8.7	41.3 ± 8.2	41.7 ± 11.8	39.8 ± 10.1
**Speaking Task**	Affectively good, bad, and neutral life memories	Affectively unspecified life memories	Affectively good, bad, and neutral life memories	-
**Speaking K and Length**	4 recordings, each 10 min long	1 recording, 5 min long	5 recordings, each 1.5 min long	-
**Speaking Task Format**	Conversation with interviewer	Conversation with interviewer	Monologue	-
**Geographic Region**	North-central US	North-central US	South-central US	-

Note. For studies where the speaking task format was a conversation with the interviewer, interviewer speech was removed before natural language processing. Age, ethnicity, and sex information were missing for 55 participants.

**Table 2 behavsci-12-00286-t002:** First 11 topics and words that constitute each topic.

Topic	Name	Emotional Valence	Top Contributing Terms
**0**	Food *	Positive	food, eat, cook, love, chicken, cooking, eating, good, favorite, lot
**1**	Hobbies *	Positive	hobbies, hobby, play, watch, love, read, just, favorite, lot, music
**2**	peace		peace, feel, relaxed, outside, just, calm, felt, really, peaceful, feeling
**3**	Routine *	Neutral	usually, work, day, habits, just, home, things, time, ah, watch
**4**	memories		know, fun, just, good, went, lot, used, memories, time, remember
**5**	people		know, people, think, ah, just, person, things, lot, time, kind
**6**	life		just, got, people, want, life, going, yes, good, know, ah
**7**	home		room, live, bedroom, living, kitchen, house, living room, nice, roommate, tv
**8**	happy		happy, best, time, just, know, good, got, award, school, real
**9**	sad		sad, felt, low, energy, feel, low energy, feel sad, passed, times, away
**10**	Scared *	Negative	know, run, bad, snake, really, got, time, going, sad, dad

* is placed next to health-related topics that passed our cosine similarity threshold. We refer the reader to Appendix A for a list of all topics.

## Data Availability

Non-identifying raw data will be considered upon request.

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
