# Peer review of "Talking about Health: A Topic Analysis of Narratives from Individuals with Schizophrenia and Other Serious Mental Illnesses"

_behavsci, 2022, doi:10.3390/bs12080286_

Round 1

Reviewer 1 Report

The article is unusual and interesting, devoted to attempts to justify the quality of life of people with schizophrenia and serious brain diseases. Individuals with schizophrenia have higher mortality and shorter lifespans. There are a multitude of factors which create these conditions, but one aspect is worse physical health, particularly cardiovascular and metabolic health. The authors talk about a potentially new way of research that includes a more patient-centered perspective; understanding the aspects of physical health that are most important and potentially most changeable for people based on their life stories. This study used topic modeling, a type of natural language processing (NLP) on unstructured speech samples from people with serious mental illness, primarily schizophrenia, to highlight topics. The article discusses the implications of these findings for how people with severe mental illness and schizophrenia think about their health, and what might be most effective for future health promotion policies and interventions. The article is well written, read with interest. This article has both scientific and practical significance.

Author Response

We thank the reviewer for their thorough and kind comments, and appreciate their time and thoughtfulness in this review! 

Reviewer 2 Report

 Dear Sir/Madam

Thanks for the practical and comprehensive article in the field of psychiatric disorders.

Author Response

We thank the reviewer for their consideration in this review. We have endeavored to improve the areas suggested. We have removed some citations from the reference list which were not immediately relevant (Michalska et al, Park et al, and Rico-Uribe et al.), and added some others to the reference list (Kidd et al. and Strauss, 2021) to address concerns about all citations being relevant, and the work being adequately referenced. In addition, we have clarified our language in the discussion, particularly in the section on health promotion, in order to address areas of improvement including arguments around findings being balanced and compelling, and thoroughly supported.  The amended section now reads as follows:

“However, one crucial nuance that our findings suggest is the importance of a positive associations with food. In a population where a loss of pleasure (i.e., anhedonia) is a cardinal symptom, the fact that food, eating, and cooking were strongly positively valenced suggests the importance of the hedonic component of food. As such, heavily restrictive diets (i.e., elimination diets) could be particularly ineffective in this population (though these diets are generally ineffective, even in populations without mental health diagnoses, McEvedy et al., 2017). Instead, advice on diet should focus on maintaining cooking and eating as a source of joy through providing culturally appropriate, non-shaming health-promoting additions to the individual’s diet.”

Reviewer 3 Report

I truly appreciate and value the authors’ work, which led to this wonderful manuscript, describing and analyzing in depth the data on how to create more effective interventions to improve the health and wellbeing of people diagnosed with schizophrenia and other serious mental illnesses. The structure of the paper is well respected and the authors paid much attention to details. “Discussion” paragraph is beautifully conceived, focusing on their results and possible explanations. The paragraph “Limitations and future directions” is also well written. Very useful manuscript! Just some minor comments:

1. Would the authors consider to add in the title also “other serious mental illnesses” ? I mean: “TALKING ABOUT HEALTH: A TOPIC ANALYSIS OF NARRATIVES FROM INDIVIDUALS WITH SCHIZOPHRENIA AND OTHER SERIOUS MENTAL ILLNESSES” (there are many patients in Study 2 and Study 3 with other serious mental illnesses – Table 1 and also some health-related topics in Results are different).

2. Abstract: I would suggest to include the study period and the number of patients, as well as the location the study was performed (I understand that patients’ data were retrieved from three previous studies of the same authors – then maybe this should be mentioned). as written in “Limitations” studies were collected across a decade and in two different parts of the United States.

3. Figure 2: Please define – in Figure legend – the abbreviation CN (controls).

4. I did not find listed in References: Kidd et al., 2016 (inserted in Page 2). Same for Strauss et al., 2021 (Page 3). Also, I did not find “Michalska da Rocha et al”, nor “Park et al” and “Rico-Uribe et al” included in the text. Please revise.

Author Response

We are grateful to the reviewer for their consideration and kind comments in reviewing this manuscript. We have included the reviewer's original comments (italicized and indented), to contextualize our point-by-point responses. 

  • Would the authors consider to add in the title also “other serious mental illnesses” ? I mean: “TALKING ABOUT HEALTH: A TOPIC ANALYSIS OF NARRATIVES FROM INDIVIDUALS WITH SCHIZOPHRENIA AND OTHER SERIOUS MENTAL ILLNESSES” (there are many patients in Study 2 and Study 3 with other serious mental illnesses – Table 1 and also some health-related topics in Results are different).

We are happy to make this change to the title. Please see revised manuscript, which has the recommended title.

  • Abstract: I would suggest to include the study period and the number of patients, as well as the location the study was performed (I understand that patients’ data were retrieved from three previous studies of the same authors – then maybe this should be mentioned). as written in “Limitations” – studies were collected across a decade and in two different parts of the United States.

We have amended the abstract, as follows, to include this information:

“…a type of Natural Language Processing (NLP) on unstructured speech samples from individuals (N = 310) with serious mental illness, primarily schizophrenia, in or-der to extract topics. Speech samples were drawn from three studies collected over a decade in two geographically distinct regions of the United States.”

  • Figure 2: Please define – in Figure legend – the abbreviation CN (controls).

Thank you for noticing this omission. We have added the definition in Figure 2. (“CN = Controls”) following the figure title.

  • I did not find listed in References: Kidd et al., 2016 (inserted in Page 2). Same for Strauss et al., 2021 (Page 3). Also, I did not find “Michalska da Rocha et al”, nor “Park et al” and “Rico-Uribe et al” included in the text. Please revise.

Thank you for noticing these mistaken references. We have removed Michalska et al, Park et al, and Rico-Uribe et al. from the reference list, and added the full references for Kidd et al and Strauss et al.

Kidd, S. A., Frederick, T., Tarasoff, L. A., Virdee, G., Lurie, S., Davidson, L., Morris, D., & McKenzie, K. (2016). Locating community among people with schizophrenia living in a diverse urban environment. American Journal of Psychiatric Rehabilitation, 19(2), 103–121. https://doi.org/10.1080/15487768.2016.1162757

Strauss, G. P. (2021). A Bioecosystem Theory of Negative Symptoms in Schizophrenia. Frontiers in Psychiatry, 12. https://doi.org/10.3389/fpsyt.2021.655471

Reviewer 4 Report

In this paper, the authors carry out a qualitative analysis of the discourse of patients with severe mental illness (schizophrenia and other diagnoses) based on audios from previous works. The aim is to obtain the factors or “topics” most highlighted by the patients. According to the authors, this information can be of help in generating more effective and efficient intervention programmes and policies.

The paper is interesting as it stands. It requires a qualitative discourse analysis, and the methodology seems adequate. However, the approach and methods raise some major doubts for me:

1.      The approach of the work (introduction and justification) seems to be established a posteriori (i.e., physical health) on the basis of the results found: main narrative topics.

2.      The authors justify the work by highlighting the social processes involved and their impact on patients' physical health, morbidity and early mortality. However, I cannot find how the social analysis is approached methodologically (I can only find a discussion of the most relevant issues in relation to the social level), can you please explain this in detail?

3.      It is not clear to me whether the general objective and the specific objectives (last paragraph of the introduction) are aimed at better explaining patients' perception of their physical and/or mental health (general health perception?). My doubt is based on the first part of the introduction where the focus seems to be on physical health.

4.      Regarding the methodology, I fail to understand the distinction between health-related and non-health-related topics (e.g., table 2. Why is "scared" a health-related topic, while "angry" is not?). I understand that the selection of emotional valence was made in the form of qualitative inspection. Also, the choice of whether a topic is health-related or non-health-related? Physical health-related, mental health-related or both? Under what specific criteria? Who or which analysis step was in charge of this inspection and selection?

5.      Also, in terms of methodology, the authors perform a thematic analysis of patients vs. controls. However, I note that there is only a record of controls only in one of the three previous works that provide narrative material. This makes it difficult, in my opinion, to compare with the totality of patients (from the discourses of the three preceding studies). Accordingly, the number of controls is finally 38 (247 speech samples) compared to 309 patients (1210 speech samples), so that the numbers are very unbalanced.

6.      Finally, and based on the results discussed by the authors, it is concluded that the main term "food" and therefore "diet" and "exercise" are key to health and should be taken into account for intervention in patients. I think that this is a very discrete conclusion and does not add much to the practical treatment of mental illness and its related physical medical problems (i.e., any professional clinician would apply these criteria).

Other minor problems include the following:

1.      The title does not reflect the focus on physical (if this is the case), rather than mental, health in the patients' narrative.

2.      Introduction, third paragraph: please give the full name of the acronym the first time it is used, not the second.

3.      Table 1 should include an additional column with overall numbers of subjects, text samples, etc.

4.      I cannot properly match the topics in Table 2 with the labels shown in Figures 1 and 2. Why does Table 2 contain 13 topics when, for example, Figure 2 highlights only 10?

Author Response

We thank the reviewer for their thoughtful and considerate review. We also appreciate the opportunity to clarify some of our methods, which undergird a number of their concerns, followed by our point-by-point responses with the reviewer's original comments (italicized and indented) for context.

While topic analysis shares some similarities with qualitative analyses, it is a quantitative analysis. Where in qualitative analyses a researcher or research team would carefully read transcripts and extract themes based on, for example, codes capturing the essence of excerpts of the text, but also informed by their own perspectives, theoretical grounding, or inquiry, topic analysis is entirely computational using natural language processing. In topic analysis, words, phrases, and entire transcripts are clustered according to an algorithm, which has been trained on a large corpus. Each word is associated with one, or more, topics based on their context within the transcripts. These topics, consisting of clusters of contextually-similar words, can then be qualitatively inspected to tag them with a human-interpretable name, much in the way one would if doing an exploratory factor analysis. We then compared each of these topics quantitatively to select health related terms, to see which topics were statistically most closely related to concepts around health. The only qualitative components of these analyses were the nicknaming of topics and the assigning of valence, which was done based on the presence of clearly valenced words (e.g. “love”, “favorite”, or “bad”). We recognize that these methods are somewhat novel, and that our description in the methods section was heavily technical. In order to increase the clarity, we have included the following description of the methods in the introduction, on page 5.

“Using topic analysis, a Natural Language Processing (NLP) method which allows for the automatic extraction of relevant topics, we aim to explore how individuals with schizophrenia perceive their own physical health behaviors as part of their larger life context. In topic analysis, the words in a transcribed speech sample are clustered according to an algorithm, which has been trained on a large corpus. These topics, consisting of contextually-similar clusters of words, can then be qualitatively assigned a name, much in the way one would if doing an exploratory factor analysis. We aimed to identify if individuals with schizophrenia discussed physical health relevant behaviors in unstructured, and untargeted speech samples, and if so, what kinds of health relevant topics emerged. In an exploratory manner, we also aimed to identify how individuals themselves valence their own physical health promoting or detrimental behaviors, and what the broader context of those behaviors are.”

The paper is interesting as it stands. It requires a qualitative discourse analysis, and the methodology seems adequate. However, the approach and methods raise some major doubts for me:

    1. The approach of the work (introduction and justification) seems to be established a posteriori (i.e., physical health) on the basis of the results found: main narrative topics.

This particular inquiry was a posteriori interested in the presence, or potential absence, of health-related topics in these speech samples. Before conducting the analyses, we were unclear as to how, or even whether, individuals with schizophrenia would discuss health related topics, but believed it would be a useful inquiry to see where they were identified by the algorithm, given the critical importance of physical health in this population.

  1. The authors justify the work by highlighting the social processes involved and their impact on patients' physical health, morbidity and early mortality. However, I cannot find how the social analysis is approached methodologically (I can only find a discussion of the most relevant issues in relation to the social level), can you please explain this in detail?

Thank you for this comment. As described above, we did not specifically analyze for social topics, or for any topics. We included this section in the introduction to provide context for how our results must be interpreted but have removed it.

  1. It is not clear to me whether the general objective and the specific objectives (last paragraph of the introduction) are aimed at better explaining patients' perception of their physical and/or mental health (general health perception?). My doubt is based on the first part of the introduction where the focus seems to be on physical health.

Thank you for identifying this lack of clarity! We’ve added the following section on page 5  to clarify the objectives.

“This study aims to continue the trajectory of centering individuals’ lived experience and perception of their health and wellbeing, but in a larger and more varied context than is feasible or applicable for qualitative inquiries. […]We aimed to identify if individuals with schizophrenia discussed physical health relevant behaviors in unstructured, and untargeted speech samples, and if so, what kinds of health relevant topics emerged. In an exploratory manner, we also aimed to identify how individuals themselves valence their own physical health promoting or detrimental behaviors, and what the broader context of those behaviors are. Lastly, we compared whether the frequency of topics differed in the group of individuals with schizophrenia, relative to other serious mental illness diagnoses or controls without a mental illness diagnosis.”

  1. Regarding the methodology, I fail to understand the distinction between health-related and non-health-related topics (e.g., table 2. Why is "scared" a health-related topic, while "angry" is not?). I understand that the selection of emotional valence was made in the form of qualitative inspection. Also, the choice of whether a topic is health-related or non-health-related? Physical health-related, mental health-related or both? Under what specific criteria? Who or which analysis step was in charge of this inspection and selection?

Thank you for highlighting this area of confusion. We have clarified the language on page 8 to show that the health-related topics were defined based on cosine similarity to health related terms.

“Second, we evaluated the number of topics that were health-related.  Topics were identified as “health-related” by evaluating cosine similarity scores between each topic and health-related terms. Cosine similarity is a method for approximating how similar two units of text (words, phrases, sentences, or documents) are using text embeddings. In brief, text embeddings are a vector (numerical) representation of a document, word, or sentence in semantic space, calculated by the probability of text appearing close to each other in natural language. Natural language is often composed oftext found on Wikipedia, social media, or other large corpora such as news articles, books, or social media. Cosine similarity scores are calculated by subtracting the cosine distance between two vectors in semantic space from one. The further the words are in semantic space (not similar), the closer their cosine similarity score is to 0. Once topics were identified, we searched resulting topics and terms that had high cosine similarity (> 0.5) to health-related terms (i.e., “health”, “exercise”, “smoking”, “alcohol”, “sleep”, and “self-care”).”

  1. Also, in terms of methodology, the authors perform a thematic analysis of patients vs. controls. However, I note that there is only a record of controls only in one of the three previous works that provide narrative material. This makes it difficult, in my opinion, to compare with the totality of patients (from the discourses of the three preceding studies). Accordingly, the number of controls is finally 38 (247 speech samples) compared to 309 patients (1210 speech samples), so that the numbers are very unbalanced.

We appreciate this concern, especially as it would be relevant in thematic analysis. Please see above for a description of the difference between quantitative topic analysis and qualitative thematic analysis.  Within a topic analysis, we were able to separate the samples by diagnostic group and extract the frequency of those topics within each group. This should address concerns of imbalanced patient-control proportions across studies.

  1. Finally, and based on the results discussed by the authors, it is concluded that the main term "food" and therefore "diet" and "exercise" are key to health and should be taken into account for intervention in patients. I think that this is a very discrete conclusion and does not add much to the practical treatment of mental illness and its related physical medical problems (i.e., any professional clinician would apply these criteria).

We appreciate the reviewer’s note on increasing the specificity of our conclusions. We wished to convey not only that diet and exercise are key to health, but that these link directly to emotional valences To more effectively convey this nuance, we have edited the text in the discussion, particularly in the “Implications for Health Promotion” subsection on page 13. The text now reads:

“However, one crucial nuance that our findings suggest is the importance of the positive associations with food. Especially in a population where a loss of pleasure, or anhedonia is a cardinal symptom, the fact that food, eating, and cooking were strongly positively valenced suggests the importance of the hedonic component of food. As such, heavily restrictive diets or elimination diets could be particularly ineffective in this population (though these diets are generally ineffective, even in populations without mental health diagnoses, McEvedy et al., 2017). Instead, advice on diet should be focused on providing culturally appropriate, non-shaming health promoting additions to diet that maintain cooking and eating as a source of joy, if it is one for the individual.”

Other minor problems include the following:

  1. The title does not reflect the focus on physical (if this is the case), rather than mental, health in the patients' narrative.

Given that the title is already quite long and another reviewer asked for additions to the title, and the abstract clearly states that this inquiry is focused on physical health, rather than change the title we have elected to add a “physical health” keyword. 

  1. Introduction, third paragraph: please give the full name of the acronym the first time it is used, not the second.

We have removed this definition of the acronym, and instead define it in the first paragraph.

  1. Table 1 should include an additional column with overall numbers of subjects, text samples, etc.

We have made the requested change to Table 1.

  1. I cannot properly match the topics in Table 2 with the labels shown in Figures 1 and 2. Why does Table 2 contain 13 topics when, for example, Figure 2 highlights only 10?

We have changed the Figure caption for Figure 2 for clarity. It now reads; Figure 2. Frequency of topic discussion by diagnostic group. For ease of interpretation, only the first eleven topics are included. Relevant topic names as seen in Table 2 are as follows: Topic 0 = Food; Topic 1 = Hobbies; Topic 2 = Peace; Topic 3 = Routine; Topic 4 = Memories; Topic 5 = People; Topic 6= Life; Topic 7 = Home; Topic 8 = Happy; Topic 9= Sad; Topic 10 = Scared; CN = Controls.

Round 2

Reviewer 4 Report

I would like to congratulate the authors for making the objectives and methodology of their work much clearer. However, in order to be able to recommend it for publication, I would like the authors to resolve some minor issues that have arisen in this latest version:

1.       The numbers given in Table 1 (310 SMI patients, 1210 language samples) do not match exactly with those reported in the text (375 patients and 1504 samples). Also, when adding the number of patients by sex (318), diagnosis (315) and ethnicity (291) they do not match either the numbers of 310 or 375 patients. Can you please check these numbers?

2.       In the section "Data analysis" I think that the explanation of the emotional value and its analysis using Chi-squared tests should be explained in a separate paragraph (e.g., the sentence "Second, we used Chi-squared test..." is confusing with respect to the first and second step of the “topics analysis” previously explained).

3.       The values on which Chi-square analyses (distribution of topics) are performed should be reported in a table per group. I think it is important to be able to see the frequency numbers/distribution group for the main topics.

4.       Table 2 now reports the first 11 topics and gains clarity. However, the text refers to other non-reported topics (15, 18). I think they should be made explicit, maybe just mentioning the topic name in the text. I would also suggest either extending Table 2 to all topics, or including the remaining topics and their main terms in a separate (perhaps supplementary) table for ease of reference.

One last question I would like to discuss with the authors: why are patients' opinions or perceptions (topics) important for the orientation of intervention programmes? The authors argue that they might bring "new insights" to be used in the intervention. However, wouldn't guidance coming from clinicians be more appropriate? Don´t you think that patients themselves might have misperceptions or misguided perceptions so that bringing them into the debate might add more noise to the search for efficient intervention strategies?

Author Response

We would like to thank the reviewer again for their continued thoughtful, considerate engagement with the manuscript. We hope to have resolved their minor concerns and believe that in doing so we have strengthened the manuscript further. Please see below for our point-by-point responses. 

  1. The numbers given in Table 1 (310 SMI patients, 1210 language samples) do not match exactly with those reported in the text (375 patients and 1504 samples). Also, when adding the number of patients by sex (318), diagnosis (315) and ethnicity (291) they do not match either the numbers of 310 or 375 patients. Can you please check these numbers.

We thank the reviewer for pointing out this oversight. The discrepancy was the result of using the “full” dataset initially but reporting sample sizes from the “cleaned and filtered” dataset for the second round. We have fixed the numbers in Table 1 and in the text.  

  1. In the section "Data analysis" I think that the explanation of the emotional value and its analysis using Chi-squared tests should be explained in a separate paragraph (e.g., the sentence "Second, we used Chi-squared test..." is confusing with respect to the first and second step of the “topics analysis” previously explained)

Thank you for highlighting this area of confusion. We edited the Data Analysis section (page 7) with the aim of making it clearer. It now reads:

We used Chi-squared tests to compare the frequency of topics between the following groups: a) Individuals with an SMI diagnosis broadly and controls, b) individuals with schizophrenia, individuals with another SMI diagnosis, and controls, and c) Black and White participants. These group comparisons were performed using a one-way Chi-squared test on the distribution across all topics. 

Topics were identified as “health-related” by evaluating cosine similarity scores between each topic and health-related terms. Cosine similarity is a method for approximating how similar two units of text (words, phrases, sentences, or documents) are using text embeddings. In brief, text embed-dings are a vector (numerical) representation of a document, word, or sentence in semantic space, calculated by the probability of text appearing close to each other in natural language. Natural language is often composed of text found on Wikipedia, social media, or other large corpora such as news articles, books, or social media. Cosine similarity scores are calculated by subtracting the cosine distance between two vectors in semantic space from one. The further the words are in semantic space (not similar), the closer their cosine similarity score is to 0. Once topics were identified, we searched resulting topics and terms that had high cosine similarity (> 0.5) to health-related terms (i.e., “health”, “exercise”, “smoking”, “alcohol”, “sleep”, and “self-care”). We assigned emotional valence to health-related topics by qualitatively inspecting the top 10 individual words in health-related topics (Table 2). We compared the frequency of health related vs all other topics among groups using a two-way Chi-square test for 7 health-related topics and all other topics.

  1. The values on which Chi-square analyses (distribution of topics) are performed should be reported in a table per group. I think it is important to be able to see the frequency numbers/distribution group for the main topics.

We understand the reviewer’s request for a contingency table and two-way chi-squared results. However, this would only be feasible for our comparison between health-related topics against all other topics. All other tests were conducted across all topics between each group and would be too large to fit into a single table. Considering this, we have put together a Supplementary Table 1, which includes all topics in addition to their frequency counts for each group. We hope you are satisfied with this addition.

  1. Table 2 now reports the first 11 topics and gains clarity. However, the text refers to other non-reported topics (15, 18). I think they should be made explicit, maybe just mentioning the topic name in the text. I would also suggest either extending Table 2 to all topics, or including the remaining topics and their main terms in a separate (perhaps supplementary) table for ease of reference.

We thank the reviewer for this suggestion. We have included a Supplementary Table S1 which includes all topics, their frequency counts, and top 10 words for each topic. Table S1 also includes “nicknames” for all topics mentioned in the text. Note that nicknames were qualitatively assigned by inspecting the top contributing words for each topic and their c-TF-ID weights. We did not feel it necessary to assign nicknames to all topics, as it is a highly subjective process.

  1. One last question I would like to discuss with the authors: why are patients' opinions or perceptions (topics) important for the orientation of intervention programmes? The authors argue that they might bring "new insights" to be used in the intervention. However, wouldn't guidance coming from clinicians be more appropriate? Don´t you think that patients themselves might have misperceptions or misguided perceptions so that bringing them into the debate might add more noise to the search for efficient intervention strategies?

We appreciate the reviewers’ thoughtful question and welcome the opportunity to address it. While we agree that a clinician’s perspective would also be appropriate (and worthy of study in its own respect), it does not preclude the importance of patient perspectives. Subjective health complaints have been associated with functional outcomes in individuals with TBI (e.g. Hofstad et al., 2016), and it is possible that similar relationships may be found in SMI, although the area is under-studied. Subjective health complaints, even if “misperceived” by the patient, can provide diagnostic information to the clinician and support more effective triage rather than simply being noise (Maeland et al., 2012). Lastly, a supportive therapeutic or treatment relationship can improve the effectiveness of any particular treatment, and treaters who dismiss subjective health complaints may damage rapport and thus limit the effectiveness of treatment. Patients are often left to feel secondary to the goals of their treatment team, and by including in a collaborative intervention development process, researchers have the ability to bridge the clinician-client gap and create future interventions that are both patient-informed and scientifically-backed. 

Hofstad, H., Næss, H., Gjelsvik, B.E.B., Eide, G.E., Skouen, J.S. (2017), Subjective health complaints predict functional outcome six months after stroke. Acta Neurologica Scandinavica, 135: 161– 169. doi: 10.1111/ane.12624 

Maeland, S., Werner, E. L., Rosendal, M., Jonsdottir, I. H., Magnussen, L. H., Ursin, H., & Eriksen, H. R. (2012). Diagnoses of Patients with Severe Subjective Health Complaints in Scandinavia: A Cross Sectional Study. ISRN Public Health2012, 851097. https://doi.org/10.5402/2012/851097